# Comprehensive Evaluation of the Properties of Ultrafine to Nanocrystalline Grade 2 Titanium Wires

**DOI:** 10.3390/ma11122522

**Published:** 2018-12-11

**Authors:** Jan Palán, Radek Procházka, Jan Džugan, Jan Nacházel, Michal Duchek, Gergely Németh, Kristián Máthis, Peter Minárik, Klaudia Horváth

**Affiliations:** 1COMTES FHT a.s., Průmyslová 995, 334 41 Dobřany, Czech Republic; radek.prochazka@comtesfht.cz (R.P.); jan.dzugan@comtesfht.cz (J.D.); jan.nachazel@comtesfht.cz (J.N.); michal.duchek@comtesfht.cz (M.D.); 2Charles University, Faculty of Mathematics and Physics, Ke Karlovu 5, 121 16 Prague, Czech Republic; gergely1227@gmail.com (G.N.); mathis@met.mff.cuni.cz (K.M.); peter.minarik@mff.cuni.cz (P.M.); horviklaudi@gmail.com (K.H.); 3Nuclear Physics Institute of the CAS, Husinec—Řež 130, 250 68 Řež, Czech Republic

**Keywords:** titanium, ECAP, Conform, continuous extrusion, wire, medical implants

## Abstract

This paper describes the mechanical properties and microstructure of commercially pure titanium (Grade 2) processed with Conform severe plastic deformation (SPD) and rotary swaging techniques. This technology enables ultrafine-grained to nanocrystalline wires to be produced in a continuous process. A comprehensive description is given of those properties which should enable straightforward implementation of the material in medical applications. Conform SPD processing has led to a dramatic refinement of the initial microstructure, producing equiaxed grains already in the first pass. The mean grain size in the transverse direction was 320 nm. Further passes did not lead to any additional appreciable grain refinement. The subsequent rotary swaging caused fine grains to become elongated. A single Conform SPD pass and subsequent rotary swaging resulted in an ultimate strength of 1060 MPa and elongation of 12%. The achieved fatigue limit was 396 MPa. This paper describes the production possibilities of ultrafine to nanocrystalline wires made of pure titanium and points out the possibility of serial production, particularly in medical implants.

## 1. Introduction

Much work has recently been done in the field of severe plastic deformation (SPD) processing of pure titanium [1]. These processes have shown great potential as they have delivered a more than two-fold improvement in mechanical properties [2,3,4,5]. With improved mechanical properties, pure titanium has become suitable for applications that require high strength levels (dental implants, bone screws, etc.). Besides that, implants with a smaller cross-section can be designed. For instance, implants with smaller load-bearing cross-sections are suitable for children [6]. In terms of biocompatibility, pure titanium is associated with increased osteoblast proliferation and better osseointegration [6,7]. Kim and Park [7] established a model explaining enhanced surface cell attachment on nanocrystallized commercially pure titanium (CP–Ti). The reason for this is that the number of nodules at the triple-point junctions of the grain microstructure increases as the grain size is refined through equal-channel angular pressing (ECAP). These properties facilitate a connection between the implant and the human tissue [6,7,8,9,10].

Commercialisation of ultrafine-grained materials prepared by SPD techniques is hindered by the fact that these techniques are rarely suitable for production outside the laboratory [1,11,12]. These techniques were generally intended for microstructural evolution observations and assessment of the properties of the investigated materials. Hence, the next step is to transfer laboratory procedures to industrial practice. Some attempts to develop processes for the continuous production of ultrafine-grained materials have taken place recently. The attempts included wire production methods such as equal-channel angular swaging (ECAS) [13] and Conform ECAP [14,15,16]. Their commercialisation is under way, particularly in the field of pure titanium wire processing. This development is associated with a research group headed by R. Z. Valiev [17], which is responsible for the development of Conform ECAP. Conform ECAP processing is a relatively new modification of the conventional ECAP technique. In this process, the principle used to generate the frictional force to push a work-piece through an ECAP die is similar to the Conform process, while a modified ECAP die design is used so that the work-piece can be repetitively processed to produce ultrafine-grained (UFG) structures. The angle of channel intersection is 120°. The input stock for this method is a square bar of 11 × 11 mm^2^ cross-sectional area [14]. To obtain the desired properties, six passes at 200–400 °C are typically carried out. The resultant semi-finished product is transferred to a wire-drawing machine, where it is worked at an elevated temperature [17,18]. This process is known as the thermomechanical treatment (TMT) of pure titanium. Wire drawing leads to intensive strengthening. Products of the Conform ECAP method have equiaxed grains with high-angle boundaries. During the subsequent drawing operation, these fine grains become elongated, their dislocation density steeply increases and the material gradually loses its plastic deformation capability. For instance, processing of Grade 4 commercially pure (CP) titanium through six passes in Conform ECAP at 200 °C and subsequent drawing at 200 °C leads to a strength of 1267 MPa [17].

An alternative for the continuous production of ultrafine-grained wires is the Conform SPD technique (Figure 1) [19]. This technique is being explored by the company COMTES FHT. Its principle is similar to that of Conform ECAP, as the feedstock is continuously forced through a chamber along an angled channel. Unlike Conform ECAP, however, Conform SPD uses round stock and the angle of its channel intersection is 90° [2,3,4,19]. The principle of forming titanium using Conform SPD technology to obtain ultrafine-grained products is shown in Figure 1. The feedstock is guided by the coining roll to the gap between the driving wheel and the shoe. High friction forces cause the feedstock to move along the groove in the driving wheel all the way to the abutment. Once the material hits the abutment, it changes direction and exits the Conform SPD machine through the chamber die [20,21,22,23,24].

This paper gives a description of the processing of CP Grade 2 titanium by a combination of Conform SPD and cold rotary swaging, and includes an overview of the final properties. It summarises information which is expected to facilitate the implementation of this material in the serial production of medical implants.

## 2. Materials and Methods

This study concerns commercially pure Grade 2 titanium (ASTM B348 Gr2). Its chemical composition is given in Table 1. The composition was determined by means of the Bruker Q4 Tasman optical emission spectrometer. The initial diameter of the wire was 10 mm. The initial material was delivered with respect to the ASTM B348 standard.

The wire was processed by the Conform SPD technology. Figure 2 shows the distribution of the strain rate, material velocity and the temperature in the die chamber, respectively. They were calculated with the DEFORM-3D software. Detailed conditions for these calculations are described in the publication [25]. The calculations described in this publication are for information only. The feedstock had a round cross-section of 10 mm diameter. At the start of the processing, it was at room temperature and no external heating was provided. Up to three passes were completed (1 pass, 2 passes, 3 passes). The driving wheel ran at 0.5 rpm. The A route was used, i.e., the feedstock orientation was identical in all passes.

In order to further increase the mechanical properties of the material, cold rotary swaging was applied in the next step. Rotary swaging was performed on the as-received material and on the material after a single pass through Conform SPD. The purpose was to compare the final properties after rotary swaging combined with Conform SPD, and after rotary swaging alone. The diameter of the initial feedstock was the same as for Conform SPD—10 mm. One operation involved the reduction of the cross-sectional area by 20%. The total reduction in cross-sectional area was up to 80%.

Metallographic specimens were prepared using a standard procedure involving grinding and subsequent polishing. The microstructure was revealed by etching with Kroll reagent. A Carl Zeiss—Observer, the Z1m optical microscope, and bright-field illumination were used for microstructure observation.

Fractographic examination was performed using the JEOL JSM 6380 scanning electron microscope. Fracture surfaces of fatigue test specimens were observed using secondary electron imaging.

Thin foils were prepared for transmission electron microscope (TEM) observations. The final electrolytic thinning was performed with the use of a Tenupol 5 device, using a solution of 300 mL CH3OH + 175 mL 2-butanol + 30 mL HClO4 at −10 °C and a voltage of 40 V. The TEM analysis was performed in a JEOL 200CX instrument, with an acceleration voltage of 200 kV. Selective electron diffraction was used for determining phases. The grain size was measured using the linear intercept method.

The electron-backscatter diffraction (EBSD) observations were performed on the cross-section of the samples using an FEI Quanta 200 FX scanning electron microscope (SEM, Thermo Fisher Scientific, Brno, Czech Republic). The EBSD measurements were conducted at a working distance of 13 mm using a step size of 50 nm at 10 kV acceleration voltage. The exact sizes of the scanned areas are indicated in the captions of particular figures. Prior to the EBSD measurement, the surfaces of the samples were ground on SiC papers (from 320 to 1200 grit) and subsequently polished for 24 h in a three-step vibratory polishing procedure. The surfaces of the specimens were finally ion-beam-polished on a Leica EM RES102 system (Leica Mikrosysteme, Wetzlar, Germany) before observation. In order to observe the microstructure of the rotary swaged sample transmission, the Kikuchi diffraction (TKD) method was used. The TKD method is also referred to as “transmission EBSD”. For ultra-fine grained materials with grain size from a few tens to hundreds of nanometers, the resolution of conventional EBSD is not sufficient since the interaction volume is comparable to the grain size. Moreover, in heavily deformed materials, EBSD analysis is obscured by high dislocation density, residual strains and lattice rotations. Transmission EBSD was measured in a Zeiss Auriga Compact FIB-SEM (Jena, Germany) using a step of 10 nm at 30 kV acceleration voltage on a standard TEM foil.

Tensile testing was carried out in an electromechanical testing machine under quasi-static loading conditions, at a constant strain rate of 0.0002/s at room temperature. Round bar tensile specimens with a diameter of 3 mm and 5 mm, respectively, were employed. Samples of 5 mm in diameter were used for wires after Conform processing. Samples of 3 mm in diameter were used for wires after Conform processing and rotary swaging. A mechanical extensometer was used for strain measurement for both testing specimen geometries.

The conventional fatigue testing procedure consists of cyclic load application on sub-sized round-bar specimens of an hourglass shape of 1.5 mm in diameter, with sinusoidal waveform at constant amplitude in the tension-compression mode. Specimens were intended to be tested under cyclic loading, with stress ratio R = −1 at room temperature. Due to the nature of bulk ultrafine-grained (UFG) materials, it was possible to use a test frequency up to 50 Hz without signs of self-heating in the high cyclic fatigue (HCF) regime verified by thermography measurements. Tests were conducted on an INOVA servo-hydraulic testing machine with a loading capacity of 15 kN. Tests were run until failure (N_f_) or up to 10 million cycles (red circles in graphs). The test results are plotted on semi-log scale in Figure 8. Due to the data scattering, a statistical approach using a 90% two-sided confidence interval was added to the data plot. The fatigue limit of 396 MPa was estimated based on two specimens surviving 10 million cycles without failure. 

## 3. Results and Discussion

### 3.1. Microstructure Evolution in Pure Titanium after Conform SPD and Rotary Swaging

The microstructure of the feedstock consisted of equiaxed recrystallized grains with annealing twins (Figure 3). These annealing twins can be seen in Figure 3a. They are a characteristic feature in materials with a hexagonal crystallographic structure due to their limited potential for deformation slip at low deformations [26]. The twins are broad and lenticular.

Figure 4 shows transmission electron micrographs of the material processed by Conform SPD with various numbers of passes. Table 2 lists average grain sizes after various numbers of passes through the Conform SPD machine. Figure 4a,b show a general view of the substructure in the transverse and longitudinal direction after the first pass. There are polyhedral grains with non-uniform dislocation density (Figure 4a) and regions with elongated grains with low-angle boundaries (Figure 4b). Variations in the grain morphology were found over the volume and the microstructure is not homogenous after the first pass. The variations in grain morphology are an indication of the non-uniform thermomechanical conditions (temperature distribution, stress state, and strain rate) within the material—the non-homogeneous character of the process is clearly visible in Figure 2.

The second and third pass produced equiaxed grains (Figure 4, from c to f). The third pass even resulted in a larger mean grain size of 420 nm in the longitudinal direction (Table 2). This increase can be attributed, in part, to non-uniform deformation and to the high surface activity of the UFG structure. High surface activity and dislocation density (strain magnitude) lower the temperatures of softening processes [4]. Certain grain growth can be expected to occur during formation due to deformation heat and the heat retained in the die chamber. This effect proves that the deformation heat (Figure 2c) is very likely to be a contributing factor in the recrystallization (post dynamic recrystallization) of the refined microstructure. The generation and effects of deformation heat on softening processes are often ignored in SPD applications. Furthermore, the decrease in grain size after the two passes is smaller than after the first pass. The refinement effect of Conform SPD seems to be limited when the initial microstructure is in the sub-micron scale [27]. A recent study [28] has shown that small grains may cause low stability when the metals are heavily deformed, and additional straining may occur as Conform SPD does not further reduce the grain size because of the intrinsic instability of nano-sized (below 100 nm) and submicrometre-sized (between 100 and 1000 nm) grains [11]. The dynamic balance of grain refinement between structure refinement and recovery at ambient temperature occurred, which was already proven in the stated study [27]. Thus, the grain refinement phenomena of Conform SPD exhibit a limit, and with additional passes, the grain size and dislocation density will reach their ultimate values. Conform SPD processing leads to substantial grain refinement and to mostly equiaxed grains in the work-piece, even in a single pass. These grains are so small that they only rotate in the subsequent passes, after which the microstructure becomes more homogeneous [29]. The mechanism by which the subgrains rotate is not so well understood. Wu et al. [11] describe a process in which dislocation motion becomes restricted due to the small subgrain size and grain rotation becomes more energetically favourable [30]. Mishra et al. [29] proposed a slightly different explanation, in which the rotation is aided by diffusion along the grain boundaries (which is much faster than through the grain interior) [29]. Conform SPD therefore reaches its grain refinement limit in just one pass. Further passes mostly lead to nothing more than homogenization of the microstructure. However, according to some studies, the maximum (limit) grain refinement is only achieved after multiple passes [11,17,18].

In order to further decrease the grain size, cold rotary swaging was employed on the material which had already been processed using the Conform device. The product of a single pass through the Conform SPD machine was rotary-swaged with 80% total reduction in the cross-sectional area. Small grains thus extended in the longitudinal direction and their dislocation density steeply increased (darker areas). This is illustrated in Figure 5.

### 3.2. Texture Evaluation

The results of the EBSD measurements conducted in the centres of samples after one, two and three Conform SPD passes are shown in Figure 6. With increasing numbers of passes, significant grain refinement occurs (please note the different scales of the images). The texture evolution is also shown in the inverse pole figure (IPF) maps in Figure 6. After the first pass, an intensive maximum can be observed close to the (0001) pole. With further passes, this maximum weakens and a new component in the inclined area between the horizontal and vertical axes is observed. This new component is dominant in the sample after three Conform SPD passes. A similar texture evolution was observed in ECAPed Mg alloys containing rare earth elements, where the intensity of this maximum was inclined by ≈ 55° from the processing direction [31]. The reason for the appearance of this component is most probably given by the activation of non-basal slip systems.

Figure 7 presents the transmission EBSD map of the sample after one SPD pass followed by rotary swaging. The microstructure, similarly to that after one SPD pass, is not homogeneous. It contains some coarse grains with some low-angle grain boundaries near the recrystallized area. The sample has a very intensive fibre texture with basal planes oriented parallel to the SPD direction. This texture is not only observed in the coarse grains but also in the recrystallized ones.

### 3.3. The Effect of Conform SPD and Rotary Swaging on Tensile Properties

Table 3 lists the tensile tests results. Materials in several differing conditions were tested after various numbers of passes through the Conform SPD machine and after rotary swaging with a reduction in cross-sectional area. The first pass through the Conform SPD machine led to an increase in Ultimate Tensile Strength (UTS) from 480 MPa for the as-received material to 580 MPa, whereas only a minimum decrease in A_5_ elongation occurred. Further passes did not bring any significant strengthening. After the second pass, the strength was 600 MPa and after the third pass it reached 623 MPa. These increases in strength were not accompanied by decreases in A_5_ elongation. These results are in agreement with the trends in grain size, since further passes caused no additional refinement and therefore there was no additional increase in strength. The cross-sectional area of a wire processed with one pass through the Conform SPD machine was further reduced by rotary swaging. It led to a notable increase in ultimate strength and yield strength. The ultimate strength was 1060 MPa, which is twice as high as in the as-received material. On the other hand, elongation was a mere 12%, in contrast to the 25% achieved with just the Conform SPD processing. This can be attributed to a much higher dislocation density and the elongation of the initially equiaxed grains [4]. The reduction in area was greater than in the initial material. This finding is in agreement with the study reported in [4]. This improvement in the reduction of area was observed for all post-Conform SPD conditions. The rotary-swaged specimen identified as “rotary swaging 80% area reduction” had a strength of 964 MPa. This is approximately 100 MPa less than in the specimen after one pass through the Conform SPD machine and rotary swaging to the same reduction level of the cross-sectional area. The elongation was 3% lower and the reduction in area was almost 24% lower in the first specimen than in the latter one. These values suggest that ductility increases for Conform SPD-processed specimens. 

### 3.4. Evaluation of Fatigue Properties

Figure 8 shows fatigue data for the specimen that underwent one pass in Conform SPD and was then rotary-swaged to an 80% reduction in cross-sectional area. This type of processing gives high values of mechanical properties while maintaining the process productivity. Its fatigue strength was 396 MPa. The fatigue strength of Grade 2 titanium in the ASTM B348 condition is approximately 240 MPa [32]. Comparing those values indicates a demonstrable increase in fatigue strength and the impact of the grain size on fatigue. Similar results were reported by other authors [32,33]. The results show that the fatigue strength of ultra-fine to nano-grained titanium at 10^7^ cycles is 60 MPa higher than conventional CP titanium Grade 4 but does not exceed that of the Ti-6Al-4V alloy, which has 530 MPa. It is clear that increasing the fatigue strength of CP titanium depends on tensile strength, this relationship being characteristic of titanium, as opposed to wavy-slip fcc materials [5]. The explanation is that the cross-slip of dislocations is more difficult in the hcp lattice. Therefore, the fatigue strength of Ti depends on the parameters of the size and shape of the grains and the type of boundaries. The twinning mechanism does not play a key role in the cyclic deformation of UFG titanium, and the fatigue mechanisms are likely related to the grain boundaries. Figure 9 shows images of the fracture surface of a ruptured fatigue test sample. It is obvious that the fatigue area covers about 60% of the fracture surface. The arrow points in the crack propagation direction. The fracture surface has been strongly smoothed out [5,27].

In summary, ultra-fine to nano-grained Ti (Grade 2) can replace conventional Ti Grade 4 with the assumption of increased component life, as the fatigue strength is roughly 60 MPa higher. The feasibility of replacing Ti-6Al-4V is still in question because its fatigue strength is about 130 MPa higher. On the other hand, the ultimate strength is about 200 MPa higher. Another advantage of commercially pure Ti is its enhanced biocompatibility as compared to the Ti-6Al-4V alloy. Recent studies, in particular, point out the toxic effects of Al and V after release of these elements into the human body. An assessment of fatigue strength of a dental implant made from ultra-fine to nano-grained titanium Grade 2 is already being evaluated by COMTES FHT and will be the subject of further articles and studies.

## 4. Conclusions

This paper presents the successful processing of commercially pure Grade 2 titanium with Conform SPD and rotary swaging techniques. The proposed route enables high-strength wires with an ultrafine to nanocrystalline microstructure to be produced in a continuous process. The article primarily describes the characteristics of the high-strength wire. The following conclusions can be stated:
Processing with Conform SPD already leads to dramatic grain refinement in the first pass. The average grain size was 320 nm. Subsequent rotary swaging further reduced the average grain size. Grains were preferentially elongated in the longitudinal direction and the sample has a very intensive fibre texture, with basal planes oriented parallel to the longitudinal direction.Combined processing with Conform SPD and rotary swaging leads to a significant increase in mechanical properties. The ultimate strength was 1060 MPa. The increase is mainly given by the refinement of the initial grain structure and by an increased dislocation density. The fatigue limit achieved for room temperature was 396 MPa.The proposed technological route gives the possibility to produce high-strength wires with an ultrafine to nanocrystaline structure. The presented paper gives an idea of how to produce such a material in a continuous way and describes the properties of the final product.

## Figures and Tables

**Figure 1 materials-11-02522-f001:**
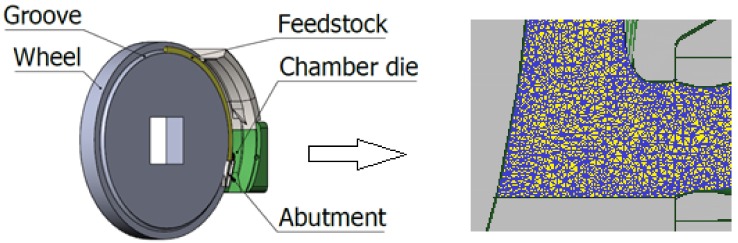
Schematic representation of the Conform severe plastic deformation (SPD) technique.

**Figure 2 materials-11-02522-f002:**
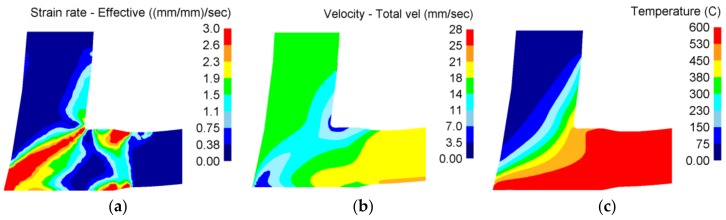
Numerical modelling of the Conform SPD process: (**a**) Strain rate distribution (s^−1^); (**b**) Velocity distribution (mm/s); (**c**) Temperature distribution (°C).

**Figure 3 materials-11-02522-f003:**
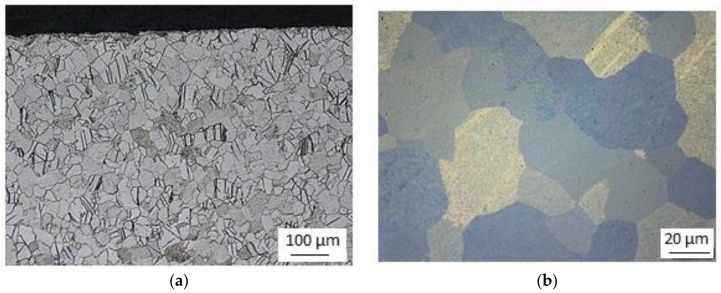
(**a**) Micrograph of the as-received structure in the transverse direction; (**b**) detailed micrograph of the as-received structure.

**Figure 4 materials-11-02522-f004:**
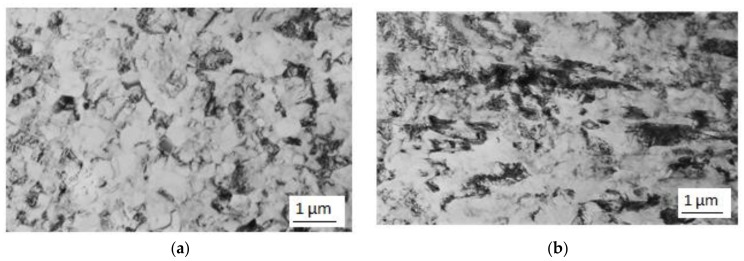
(**a**) Substructure in the transverse direction after the first pass; (**b**) substructure in the longitudinal direction after the first pass; (**c**) substructure in the transverse direction after the second pass; (**d**) substructure in the longitudinal direction after the second pass; (**e**) substructure in the transverse direction after the third pass; (**f**) substructure in the longitudinal direction after the third pass.

**Figure 5 materials-11-02522-f005:**
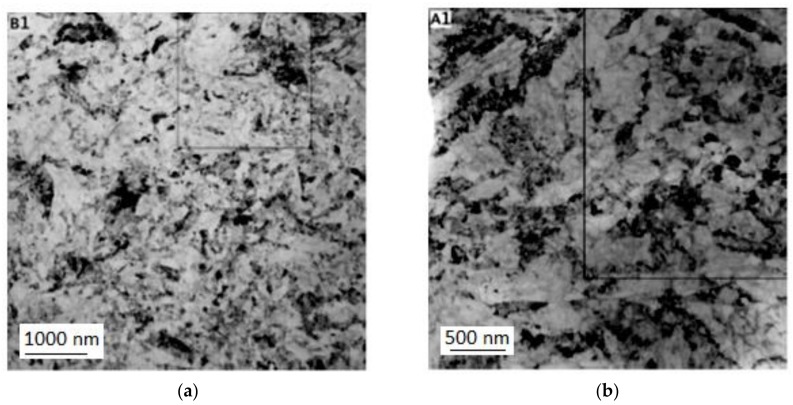
(**a**) Substructure in the longitudinal direction after one pass through the Conform SPD machine and rotary swaging; (**b**) substructure in the longitudinal direction after one pass and rotary swaging.

**Figure 6 materials-11-02522-f006:**
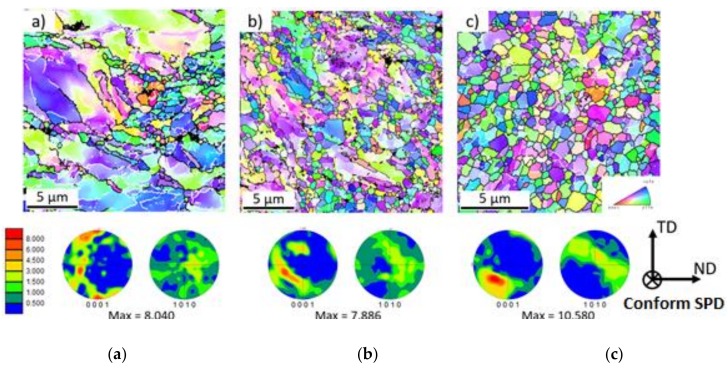
Cross-section orientation image maps (OIM) and inverse pole figure (IPF) maps for samples after (**a**) first pass (50 × 50 µm), (**b**) second pass (20 × 20 µm), and (**c**) third pass (15 × 15 µm) through Conform SPD. In the figure on the left, the scale of texture intensities is shown as multiples of the random density (m.r.d.) from 0–8.000. The maximum value of each texture is listed below the IPF maps. The orientation triangle for the electron-backscatter diffraction (EBSD) maps is shown in the right-hand corner of the figure.

**Figure 7 materials-11-02522-f007:**
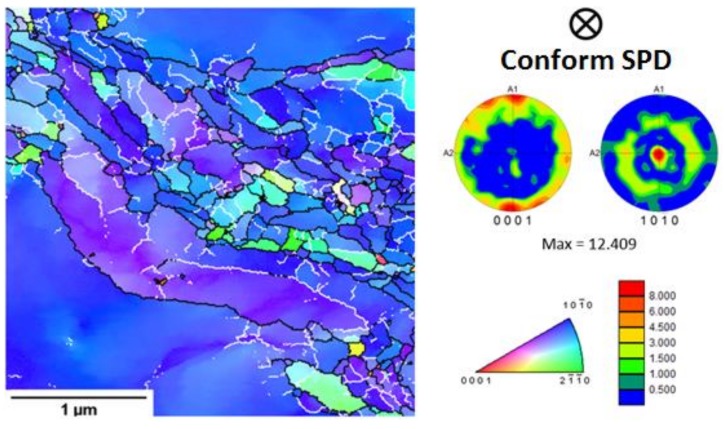
Transmission EBSD map of the sample after one Conform SPD pass and rotary swaging (3 × 3 µm). High-angle grain boundaries (>15°) are marked by black lines, while the low-angle grain boundaries (between 4° and 15°) are marked by white lines. The texture intensities are carried out as the multiples of random density (m.r.d.) from 0–8.000, with maximum at 12.409 m.r.d.

**Figure 8 materials-11-02522-f008:**
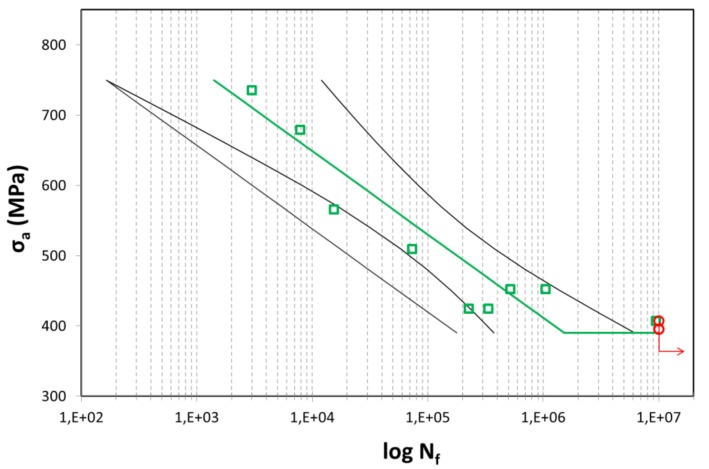
S–N curve for the sample after Conform SPD one pass + rotary swaging (80% area reduction).

**Figure 9 materials-11-02522-f009:**
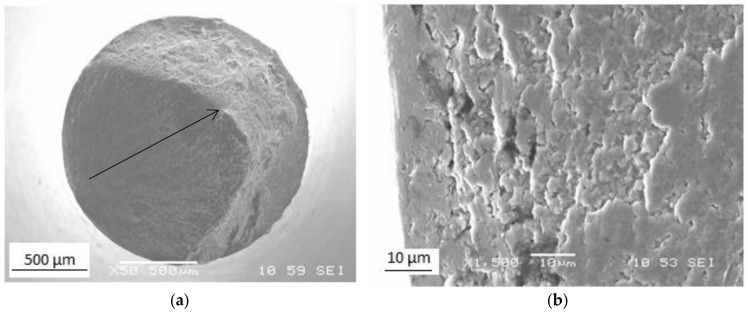
Fracture surface of the sample (Conform SPD one pass + rotary swaging (80% area reduction)) after (**a**) fatigue testing and (**b**) fracture initiation area.

**Table 1 materials-11-02522-t001:** Chemical composition of feedstock, wt. %.

Element	Fe	O	C	H	N	Ti
Content	0.046	0.12	0.023	0.0026	0.0076	balance

**Table 2 materials-11-02522-t002:** Mean grain size for different processing steps.

	Transverse Direction	Longitudinal Direction
As received	28.95 µm	
1 pass	320 ± 35 nm	310 ± 30 nm
2 passes	250 ± 25 nm	310 ± 30 nm
3 passes	330 ± 30 nm	420 ± 30 nm

**Table 3 materials-11-02522-t003:** Mechanical properties after Conform SPD processing and after Conform SPD + rotary swaging. Ultimate tensile strength (UTS); offset yield (OYS); reduction in area (RA); elongation (A_5_).

Condition	0.2 OYS [MPa]	UTS [MPa]	A_5_ [%]	RA [%]
As received	370 ± 9.4	480 ± 7.7	25 ± 1.3	52 ± 1.9
Conform SPD—1 pass	540 ± 5.8	580 ± 6.1	23 ± 1.2	62 ± 2.3
Conform SPD—2 passes	560 ± 1.6	600 ± 5.6	23 ± 1.3	62 ± 2.2
Conform SPD—3 passes	570 ± 1.8	623 ± 4.8	23 ± 1.4	62 ± 2.1
Conform SPD 1 pass + Rotary Swaging (80% area reduction)	975 ± 2.3	1060 ± 4.6	12 ± 1.4	58 ± 2.3
Rotary Swaging (80% area reduction)	890 ± 2.1	964 ± 4.4	9 ± 1.3	34.2 ± 2.4

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
