# Peer review of "Comprehensive Evaluation of the Properties of Ultrafine to Nanocrystalline Grade 2 Titanium Wires"

_materials, 2018, doi:10.3390/ma11122522_

Reviewer 1 Report

The authors describe a method to do serial production  of ultrafine nanocrystalline wires of pure titanium. This structures are experimental well investigated as well as the results are simulated. Since the work has a real possibility for production I recommend publication. However, the authors should  polish the text. Some examples are: line 44 (innvestigated), line 256 (Table1.), etc

Author Response

Author's Reply to the Review Report (Reviewer 1)

Please find corrections below:

Many changes in the text concerned definite and indefinite articles. Comments on them are not included here.

Innvestigated -> investigated – typographical error, corrected

Word order in the title of the paper has been changed to remove an unnecessary “of” structure

Further passes have not led to additional appreciable grain refinement. -> Further passes did not lead to any additional appreciable grain refinement. – the present perfect tense structure was removed because the outcome of the action (grain refinement) was altered by subsequent process steps, and therefore simple past tense is more appropriate

points out serial production possibility in, particularly for medical implants -> points out the possibility of serial production in particularly medical implants – the expression “serial production possibility” was changed to a structure with “of” to improve readability; the redundant preposition “in” was removed and the reference to medical implants corrected

Commercionally Pure Titanium -> Commercially Pure Titanium – an incorrect form of adverb was replaced with the correct form

responsible for the Conform ECAP development -> responsible for the development of Conform ECAP – word order changed for better clarity

This technique is explored by the company COMTES FHT. -> This technique is being explored by the company COMTES FHT. – Present continuous tense is more appropriate for describing the company’s ongoing activity.

This study concerns with  ->  This study concerns – Incorrect verb-preposition structure was corrected.

with a respect to ->  with respect to – Indefinite article removed.

was used for phases determination -> was used for determining phases – A noun-based structure, which is common in Czech and German, for instance, was replaced with a verb-based expression.

was rotary-swagged -> was rotary-swaged – The verb form was corrected.

Results and discusion -> Results and discussion – Typographical error corrected.

These annealing twins can be seen in Figure 3 b. -> These annealing twins can be seen in Figure 3 a. – The reference to a micrograph with twins was updated.

Figure 2 ->  Figure 3 – The number of the figure was corrected.

This effect proves the deformation heat (Figure 2 c) which is very likely contributing factor to the recrystallization (post dynamic recrystallization) of the refined microstructure. -> This effect proves the deformation heat (Figure 2 c) is very likely a contributing factor in the recrystallization (post dynamic recrystallization) of the refined microstructure. – The word “which” was removed to make the sentence clear.

The mechanism by which the subgrains rotate is less understood. -> The mechanism by which the subgrains rotate is not so well understood. – The expression was changed to read more natural in English.

and grain rotation becomes more energetically favorable -> and grain rotation becomes more energetically favourable – The U.S. English form of the word “favorable” was changed to the British “favourable”.

appereance  -> appearance – Typographical error corrected.

In order to further decrease the grain size, cold rotary swaging was employed on material which was already processed using the Conform device. -> In order to further decrease the grain size, cold rotary swaging was employed on material which had already been processed using the Conform device. – Simple past tense in the last clause was changed to past perfect tense in order to reflect the sequence of actions in this sentence.

fiber  -> fibre – Updated to British English.

with maximum ar 12.409 m.r.d.  -> with maximum at 12.409 m.r.d. – Typographical error (“ar”) corrected.

Table 1  -> Table 3 – The numbering of tables was corrected.

Table 3: commas as decimal separators were replaced with decimal points.

There is a clear fact that increasing of the fatigue strength of CP titanium depends on tensile strength, that is a feature of titanium as opposed to fcc wavy slipmaterials [5]. This would be explained with difficulty of dislocation cross slip in the HCP lattice.  ->  It is a clear fact that increasing the fatigue strength of CP titanium depends on tensile strength, this relationship being characteristic of titanium, as opposed to wavy-slip fcc materials. The explanation is that cross slip of dislocations is more difficult in hcp lattice. – The sentences were reworded for better clarity.

Replacement of Ti-6Al-4V is still a question as the fatigue life is about 130 MPa lower.  -> The feasibility of replacing Ti-6Al-4V is still a question because its fatigue strength is about 130 MPa higher. – The sentence was reworded for better clarity.

fatigue life  -> fatigue strength – Since the values are give in MPa, the term “life” was replaced with the appropriate term “strength”.

4. Conclusions -> 5. Conclusions – The number of the last section was corrected.

Reviewer 2 Report

In this paper, authors studied the mechanical properties and microstructure evolution of commercially pure titanium (Grade 2) processed with Conform SPD and rotary swaging techniques. The authors have shown that Combined processing with Conform SPD and rotary swaging has led to a significant increase in mechanical properties. The manuscript is a well-organized good paper. The topic of this article is of high interest to researchers.

I propose some minor amendments:

1.     Line 81: «This study concerns with commercially pure Grade 2 titanium (ASTM B348 Gr2). Its chemical composition is given in Table.» Missing table number.

2.     Error in the numbering of pictures. Figure 4 after Figure 2.

3.     Figure 4.Scale marks are very small. I think you need to increase the font size. For example, as in Figure 2.

4.     Figure 7:Invisible grain boundaries. I think, that authors should make grain boundaries thicker.

5.     Figure 6:The grain boundaries in this figure are also not visible. Authors need to increase the thickness of the grain boundaries. Moreover, it is better to indicate the designation of high-angle and low-angle boundaries in the figure.

Line 256: Wrong table number.

Author Response

Author's Reply to the Review Report (Reviewer 2)

Please find corrections bellow:

1.The missing number in Table 1. was filled.

2.Number of the Figure 3. was corrected.

3.Scales in Figure 4. were corrected.

4. Grain boundaries were highlighted in Figure 7.

5. Grain boundaries were highlighted in Figure 6.

6. The caption of Table 2. was corrected.

7. Grain boundaries were highlighted in Figure 6

Reviewer 3 Report

General comments: The article demonstrated the continuous processing of CP grade 2 titanium to refine grain sizes from ultrafine size to nano-crystalline sizes. The SPD microstructures and the corresponding texture evolution are accurately presented by TEM and EBSD investigations. While some of the mechanical properties are previously reported in ref. 4, the present paper additionally identifies interesting fatigue properties for the rotary swaged sample following conform SPD process. The discussion of the results is poorly written.

Comments:

Figure 3 was incorrectly captioned as Figure 2. The quality of the optical micrographs is poor. Please provide better quality images.

In the case of rotary swaged sample in line 260, the authors stated “This type of processing gives high level of mechanical properties while maintaining the process productivity”. More details needs to be given what is the ‘high level’ mechanical properties means? What is the desired properties for medical implant applications? And how does this particular sample processing (conform SPD + rotary swaging) over conform SPD is beneficial for ‘maintaining process productivity’.

What does the legend (Exp. data d1,5) in Figure 8 implies?

The discussion relating the fatigue properties is incomplete. Please discuss related literature of fatigue properties for medical applications.

Author Response

Author's Reply to the Review Report (Reviewer 3)

Please find corrections bellow:

1. Number of the Figure 3. was corrected.

2. Optical micrographs (Figure 3.) were replaced.

3.The discussion related to fatigue behaviour has been extended:

Figure 8 shows fatigue data for the specimen which underwent one pass in Conform SPD and then Figure 8 shows fatigue data for the specimen which underwent one pass in Conform SPD and was then rotary-swagged to an 80% reduction in cross-sectional area. This type of processing gives high values of mechanical properties while maintaining the process productivity. Its fatigue lifestrength was 396 MPa. The fatigue strengthlife of Grade 2 titanium in the ASTM B348 condition is approximately 240 MPa [32]. Comparingson between those values indicates a demonstrable increase in fatigue strengthlife and the impact of the grain size on fatigue. Similar results were reported by other authors [32, 33]. RThe results shows that the fatigue strength of ultra- fine to nano- grained titanium at 107 cycles is 60 MPa higher than conventional CP titanium grade 4 but does not exceed that of the Ti-6Al-4V alloy, which has 530 MPa. ThereIt is a clear fact that increasing of the fatigue strength of CP titanium depends on tensile strength, thatwhich is a featurethis relationship being characteristic of titanium, as opposed to fcc wavy -slip fcc materials [5]. This would bee explainednation with difficulty of dislocationis that cross slip of dislocations is more difficult in the HCPhcp lattice. Therefore, the fatigue strengthlife of Ti depends on the parameters of the size and shape of the grains and the type of the boundaries. The twinning mechanism does not play a key role in the cyclic deformation of UFG titanium, and the fatigue mechanisms are likely related to the grain boundaries. Figure 9 shows images of the fracture surface of a ruptured fatigue test sample. It is obvious that the fatigue area covers about 60% of the fracture surface. The arrow points in the crack propagation direction. The fracture surface has been strongly smoothed out [5, 27].

In summary, the ultra- fine to nano- grained Ti (Grade 2) can replace the conventional Ti Garade 4 with the assumption of increased component life, as the fatigue strengthlife is roughly 60 MPa higher. The feasibility of Rreplacement ofing Ti-6Al-4V is still a question as thebecause is fatigue strengthlife is about 130  MPa lowerhigher. On the other hand, the ultimate strength is about 200 MPa higher. Another advantage of commercially pure Ti is its enhanced biocompatibility as compared to Ti-6Al-4V alloy. Recent studies, in particular, point out the toxic effects of Al and V after releaseing of these elements into the human body. An assessment of fatigue strengthlife of thea dental implant made from ultra-nano fine to nano- grained titanium grade 2 is already being evaluated by COMTES FHT and will be the subject of further articles and studies.

4. Figure 8. was corrected

Round  2

Reviewer 3 Report

The discussion related to fatigue is improved. Now the manuscript highlightied the importance of CP Ti over Ti64 for application in bio implants.